# The Diagnostic Values of Peptidoglycan, Lipopolysaccharide, and (1,3)-Beta-D-Glucan in Patients with Suspected Bloodstream Infection: A Single Center, Prospective Study

**DOI:** 10.3390/diagnostics12061461

**Published:** 2022-06-14

**Authors:** Ying Zhao, Ze-Yu Wang, Xue-Dong Zhang, Yao Wang, Wen-Hang Yang, Ying-Chun Xu

**Affiliations:** 1Department of Clinical Laboratory, Peking Union Medical College Hospital, Peking Union Medical College, Chinese Academy of Medical Sciences, Beijing 100730, China; yaoo_wang@163.com (Y.W.); yangwenhang@139.com (W.-H.Y.); 2Beijing Key Laboratory for Mechanisms Research and Precision Diagnosis of Invasive Fungal Diseases, Beijing 100730, China; 3Department of Clinical Research, Autobio Diagnostics Co., Ltd., No. 87 Jingbei Yi Road, National Eco&Tech Zone, Zhengzhou 450000, China; develop@autobio.com.cn; 4Department of Clinical Laboratory, The First Affiliated Hospital of Xi’an Jiao Tong University, No. 277 West Yanta Road, Xi’an 710061, China; zhangxuedong@autobio.com.cn

**Keywords:** bloodstream infection, peptidoglycan, lipopolysaccharide, (1,3)-Beta-D-Glucan, diagnostic efficacy

## Abstract

This study aimed to assess the diagnostic values of peptidoglycan (PGN), lipopolysaccharide (LPS) and (1,3)-Beta-D-Glucan (BDG) in patients with suspected bloodstream infection. We collected 493 heparin anticoagulant samples from patients undergoing blood culture in Peking Union Medical College Hospital from November 2020 to March 2021. The PGN, LPS, and BDG in the plasma were detected using an automatic enzyme labeling analyzer, GLP-F300. The diagnostic efficacy for PGN, LPS, and BDG were assessed by calculating the sensitivity, specificity, positive predictive value (PPV), and negative predictive value (NPV). This study validated that not only common bacteria and fungi, but also some rare bacteria and fungi, could be detected by testing the PGN, LPS, and BDG, in the plasma. The sensitivity, specificity, and total coincidence rate were 83.3%, 95.6%, and 94.5% for PGN; 77.9%, 95.1%, and 92.1% for LPS; and 83.8%, 96.9%, and 95.9% for BDG, respectively, which were consistent with the clinical diagnosis. The positive rates for PGN, LPS, and BDG and the multi-marker detection approach for PGN, LPS, and BDG individually were 11.16%, 17.65%, and 9.13%, and 32.86% significantly higher than that of the blood culture (*p* < 0.05). The AUC values for PGN, LPS, and BDG were 0.881 (0.814–0.948), 0.871 (0.816–0.925), and 0.897 (0.825–0.969), separately, which were higher than that of C-reactive protein (0.594 [0.530–0.659]) and procalcitonin (0.648 [0.587–0.708]). Plasma PGN, LPS, and BDG performs well in the early diagnosis of bloodstream infections caused by Gram-positive and Gram-negative bacterial and fungal pathogens.

## 1. Introduction

Bloodstream infections (BSIs) are systemic infectious diseases that could lead to bacteremia, sepsis, shock, disseminated intravascular coagulation, and multiple organ failure [1]. In recent years, with the increase of invasive operation and the irrational application of broad-spectrum antibiotics and corticosteroids, the morbidity and mortality of BSI has increased year by year [2,3]. Although blood culture is still the gold standard for diagnosing bloodstream infections (BSIs), this method has limitations for judging the patient’s condition timely and accurately, owing to the low sensitivity and reporting delay [4,5]. Therefore, early diagnosis of patients with BSI is vital, but remains a major challenge.

Currently, inflammatory markers such as interleukin 6 (IL-6), C-reactive protein (CRP), and procalcitonin (PCT) are commonly used in the early diagnosis of BSIs [6]. However, these inflammatory factors cannot distinguish the microorganisms of infection, and fail to reduce the unnecessary use of antibiotics. Thus, it is urgent and important to select appropriate biomarkers in order to quickly distinguish different pathogens and take appropriate treatment strategies to reduce the mortality of BSIs.

Lipopolysaccharide (LPS) is one of the components of the cell wall of Gram-negative bacteria, which could activate inflammatory cells and inflammatory factors in tissue, then lead to fever and systemic inflammatory response (SIRS), and even cause severe pathophysiological symptoms such as disseminated intravascular coagulation, shock or multiple organ failure. The level of LPS in body fluid is an important specific indicator for severe bacterial inflammation caused by Gram-negative bacteria [7,8]. In recent years, the number of patients with BSIs caused by Gram-positive bacteria has been increasing [9]. Peptidoglycan (PGN) is a part of the bacterial cell wall, accounting for about 70% of Gram-positive bacteria. The phenoloxidase activation system, an important part of the non-specific immune system, could be specifically activated by PGN to cause a series of cascade reactions and finally produce melanin [10,11,12]. (1,3)-Beta-D-Glucan (BDG), accounting for more than 50% of the fungal cell wall, is a structural component of the cell wall of most fungi. After fungi enters the human blood or deep tissue, BDG is released from the cell wall, and is increased in the blood and other body fluids. Therefore, BDG could be utilized for the early diagnosis of invasive fungal disease (IFD) and for evaluation of the clinical treatment effect [13,14].

BDG as a unique component of the fungal cell wall structure has been widely used in the early diagnosis of invasive fungal infections [15]. LPS is an important indicator for the diagnosis and monitoring of Gram-negative bacterial infections [16]. At present, many manufacturers have relevant products detecting LPS and BDG for the clinical diagnosis of Gram-negative bacterial infections and IFI. Although the study of [17] showed that PGN can be used for the detection of bacterial infection, studies of PGN regarding the source of pathogenic microorganisms in blood are still relatively rare and there are no relevant products; therefore, we developed PGN detection products and studied the multi-marker detection.

In this study, we adopted the research method of large samples; took blood culture as the main comparative methodology; and comprehensively analyzed the results of the body temperature records, CRP, PCT, white blood cell (WBC), blood culture, microscopic examination, and clinical outcomes to evaluate the clinical diagnostic efficacy of the multi-marker detection approach in patients with suspected BSIs.

In conclusion, our research indicated that the multi-marker detection approach could effectively help in the rapid diagnosis of BSIs, provide references for the rational use of antimicrobial agents, and improve the prognosis of the disease. The multi-marker detection approach was expected to be applied to the clinical diagnosis, and could provide new options for the accurate and rapid diagnosis of BSIs.

## 2. Materials and Methods

### 2.1. Samples

We collected 493 heparin anticoagulant samples from patients with suspected BSI who were undergoing blood culture in Peking Union Medical College Hospital from November 2020 to March 2021. Samples collected from the same patient more than once were only included in the group once; moreover, the sample collection time was within 24 h from the blood culture sample collection time. Samples with serious lipaemia, hemolysis, turbidity, and microbial contamination, or samples not stored according to the requirements of the reagent instructions were excluded.

### 2.2. Instruments and Reagents

The BACT/ALERT VIRTUO blood culture system (BioMérieux, Marseille-L’Étoile, France), Bactec FX blood culture system (Becton-Dickinson, Sparks, MD, USA), and their supporting blood culture bottles were used for blood culture. The Vitek 2 Compact system (BioMérieux, Marseille-L’Étoile, France) and Autof ms1000 (Matrix-Assisted Laser Desorption/Ionization Time of Flight Mass Spectrometry) (Autobio Diagnostics, Zhengzhou, China) were used for the identification of bacteria and fungi. The markers of PGN, LPS, and BDG were measured by Gram-positive bacterium Peptidoglycan Determination Assay (Chromogenic) (Approval number: 20192400460), Gram-negative Bacterial Endotoxin Chromogenic Assay (Approval number: 20192400268), Fungus (1,3)-β-D-glucan Chromogenic Assay (20192400281), and the GLP-F300 automatic enzyme labeling analyzer (Approval number: 20202221003) (Autobio Diagnostics, Zhengzhou, China).

### 2.3. Methods

Venous blood was extracted using a heparin anticoagulant tube according to scale and was centrifuged at 3000 rpm for 10 min for reserve use.

The concentrations of PGN, LPS, and BDG in the heparin anticoagulant plasma samples were detected strictly according to the reagent instructions and instrument operation manual, respectively, and the results were subject to the instrument report. The interpretation criteria for the samples of patients with bacterial bloodstream infection were as follows: first, combined with clinical treatment conditions, patients with positive blood culture results excluding contamination were considered positive. In addition, samples with negative blood culture results but positive detection kits (Gram-positive bacterium Peptidoglycan Determination Assay, Gram-negative Bacterial Endotoxin Chromogenic Assay, and Fungus (1,3)-β-D-glucan Chromogenic Assay), as long as one or more of the following indexes were met, the results were judged as being positive: ➀ Patients clinically diagnosed as having sepsis and the corresponding antimicrobial treatments were effective. ➁ Patients obtained positive bacterial culture results in other sterile body fluids such as cerebrospinal fluid, pleural effusion, drainage fluid, or puncture fluid. Finally, samples not meeting the judgment criteria mentioned above were judged as negative.

The interpretation criteria for the samples of patients with BSIs caused by fungi were as follows: The results of the blood culture were positive. Microscopic examination or culture results of the sterile tissue samples and sterile body fluids (excluding urine, bronchoalveolar lavage fluids, and cranial sinus cavity) were positive. Pneumocystis jiroveciiascus (cysts containing ascospores) and trophic forms were found in the lung tissue samples, bronchoalveolar lavage fluid, or sputum using Giemsa and Gomori’s methenamine silver (GMG) staining. The test results of the pneumocystis nucleic acid were positive. Samples reached the level of probable diagnosis of IFD according to the Consensus Definitions of Invasive Fungal Disease [18], although without definite evidence, we classified those samples as positive samples. Moreover, the samples with the results of the blood culture or other sterile body fluid culture negative; the indexes of infection were normal or with mild local infections were negative.

### 2.4. Statistical Analysis

The results of the detection kits and blood culture were counted in order to compare the coincidence rate. The contingency tables were used for summarizing the data and calculating the sensitivity, specificity, positive predictive value (PPV), negative predictive value (NPV), total coincidence rate, and 95% confidence interval (CI) of the detection kits. The Kappa test was used for analyzing the consistency between the detection kits and the comparison method. SPSS statistical software (version 28.0, SPSS Inc., Chicago, IL, USA) was used for analyzing the general information for patients, Graph Pad 7.0 was used to calculate the difference between the multi-marker detection approach and CRP, and PCT was used for the diagnosis efficacy of BSIs.

## 3. Results

### 3.1. General Information for Patients

A total of 493 patients, aged 0–94 years, were included from November 2020 to March 2021. As shown in Table 1, there were significant differences in age from three markers or blood culture plus clinical diagnosis groups, indicating that old age was also one of the susceptibility factors for bloodstream infection, while gender had no significant difference.

### 3.2. Strain Identification in Patients with BSIs

This research showed that the most frequently isolated Gram-negative bacteria were Klebsiella pneumonia (34.9%) and Escherichia coli (28.6%) in peripheral blood, and Klebsiella pneumonia (23.5%) and Escherichia coli (23.5%) in sterile body fluids and other sterile tissues. The most frequently isolated Gram-positive bacteria were Enterococcus faecium (20%), Staphylococcus aureus (15%), and Staphylococcus epidermis (15%) in peripheral blood, and Enterococcus faecium (45.5%) in sterile body fluids and other sterile tissues. The most frequently isolated fungi was Candida albicans (50%) in peripheral blood samples of patients with BSI, which is consistent with previous publications [19]. Pneumocystis jirovecii (69.6%) was the main pathogen found in sterile body fluids and other sterile tissues, which was significantly different from the common fungal pathogen in BSIs. In addition, except for common species, some rare bacteria and fungi could also be detected using our test (See Appendix A for details).

### 3.3. The Results of the Multi-Marker Detection Approach of LPS, PGN, and BDG in the Diagnosis Were Consistent with the Clinical Diagnosis in BSIs

In this study, 493 clinical samples were detected. There were 50 Gram-positive bacteria positive cases, including 40 cases of peripheral blood culture and 10 cases of sterile body fluid culture; among these, there were 8 cases of positive blood culture but with a negative PGN test, which were clinically considered as contamination, and the patients improved without corresponding treatment, so they were not counted as positive BSIs samples, combined with their clinical diagnosis. As a detection biomarker of BSIs, compared with the blood culture combined with the clinical diagnosis, the sensitivity, specificity, and coincidence rate of PGN were 83.3%, 95.6%, and 94.5%, respectively. There were 86 Gram-negative bacteria positive cases, including 60 cases of peripheral blood culture and 16 cases of sterile body fluid culture (one of them was also positive for peripheral blood culture), and another 11 cases with severe pulmonary infection, which were considered as positive from the results of the bacterial culture in the sputum or alveolar lavage fluid and which corresponded to clinical treatment. The sensitivity, specificity, and coincidence rate of LPS were 77.9%, 95.1%, and 92.1%, respectively, compared with the blood culture combined with the clinical diagnosis. There were 37 positive cases of fungi, including 8 cases of peripheral blood culture; 6 cases of sterile body fluid culture; 16 cases for Pneumocystis jirovecii pneumonia (PCP), which were positive upon microscopic examination or nucleic acid detection; and other 7 positive cases meeting the diagnostic criteria of probable IFD. Among them, two cases were *Candida albicans* positive in the blood culture, but had a negative BDG test, and combined with the clinical analysis, we thought the reason for this was that the catheter-tip bacteria caused by a long-term indwelling catheter disseminated into the body, but were quickly eliminated by the immune system, so they failed to form a continuous infection. There was one case where the blood culture was *Cryptococcus neoformans*, but it obtained a positive result for BDG, which was inconsistent with what was described in the literature, that the main components of the *Cryptococcus* cell wall were (1-6)- β-D-glucan, and that a thick capsule on the outside of the cell wall may inhibit the release of its glucan and so *Cryptococcus* infection may not obtain a positive result for BDG [20]. Combined with the clinical analysis, in this case, the patient had an HIV infection, was clinically considered as having PCP, and was treated with sulfa drugs. It was speculated that the positive result was caused by PCP.

As a detection biomarker of IFD, the sensitivity, specificity, and coincidence rate of BDG were 83.8%, 96.9%, and 95.9%, respectively. Moreover, the results in this research showed that the sensitivity, specificity, and coincidence rate of the multi-marker detection approach were 81.1%, 88.7%, and 86.4%, respectively, which were highly consistent with the results of the blood culture combined with the clinical diagnosis. The specific results are shown in Table 2 and Table 3.

### 3.4. The Positive Rates of PGN, LPS and BDG Were Significantly Higher Than That of Blood Culture

The results in Table 4 suggest that the positive rates of PGN, LPS, and BDG and the multi-marker detection approach were 11.16%, 17.65%, 9.13%, and 32.86%, respectively, significantly higher than the 6.49%, 12.17%, 1.62%, and 19.88%, respectively, in the blood culture (χ^2^ = 6.669, 5.828, 27.297, and 21.396, respectively, *p* < 0.05).

### 3.5. The Diagnostic Efficacy of PGN, LPS, and BDG in BSIs Was Better Than That of CRP and PCT

In this research, among the 493 samples, 376 cases had PCT results and 354 cases had CRP results. The positive rate of PCT was judged according to different cut off values, as shown in Table 5. The ROC curves of PGN, LPS, BDG, CRP, and PCT were analyzed using spass22.0 software (Figure 1). The results in Table 6 show that the AUCs under the ROC curves of PGN, LPS, BDG, CRP, and PCT were 0.881 (0.814–0.948), 0.871 (0.816–0.925), 0.897 (0.825–0.969), 0.594 (0.530–0.659), and 0.648 (0.587–0.708), respectively, and these biomarkers had certain values in the complementary diagnostics of BSI. The areas under the ROC curve of PGN, LPS, and BDG were significantly higher than that of CRP and PCT (Figure 1), which had better diagnostic values for BSIs.

## 4. Discussion

Bloodstream infections, with a high morbidity and mortality, are becoming an increasingly serious public health problem [1]. Currently, blood culture is still the gold standard for the diagnosis of BSIs. Although the improvement in the continuous blood culture system has greatly shortened the reporting time of the blood culture, the usual positive reporting time is still concentrated at 13.5 h–19.3 h [21]. Moreover, the positive rate is still low for actual clinical infection, and there are lots of false positives caused by contamination in BSIs [22,23]. The early diagnosis and treatment of BSIs are very important for patient prognosis; therefore, the rapid and accurate identification of pathogenic microorganisms is of great significance for clinical treatment in BSIs.

CRP and PCT are common infection biomarkers in clinic, but they also have some limitations. For example, the level of CRP is also increased in patients with tissue injury, virus infection, or slight local infection, but does not reach the level of BSIs. In addition, the level of PCT is also nonspecifically increased after surgery, multiple trauma, autoimmune system diseases, insufficient tissue perfusion, and the interference of some drugs [24,25]. PGN, LPS, and BDG as the main components of the cell wall of Gram-positive bacteria, Gram-negative bacteria, and fungi, respectively, could be used as effective biomarkers for the early diagnosis of BSIs caused by bacteria and fungi [7,9,12].

In this study, the positive rate of PCT in samples with a positive blood culture combined with clinical diagnosis was 21.0%, which was significantly lower than that of the multi-marker detection approach (81.58%). At the same time, the AUC and ROC curves of PGN, LPS, and BDG were higher than that of CRP and PCT, indicating that the efficacy of PGN, LPS, and BDG in the diagnosis of BSIs was better than CRP and PCT. In addition, compared with blood culture combined with clinical diagnosis, the coincidence rates of PGN, LPS, BDG, and the multi-marker detection approach were 94.5%, 92.1%, 95.9%, and 86.4%, respectively, which were highly consistent with the results of the clinical diagnosis. The multi-marker detection approach, as a joint screening project for the early diagnosis of BSIs, could quickly assist clinicians in judging the pathogens of infection within three hours, and then provide references for the clinical use of antibiotics.

Blood culture contamination caused by incomplete skin disinfection and improper blood collection has often been regarded as a problem. Paxton A pointed out that about 20–50% of positive samples of blood culture are contaminated [26] and *coagulase negative staphylococcus* accounts for 70–80% of the common blood culture contaminated bacteria [27]. The misjudgment of contaminated bacteria could lead to the overuse of antibiotics, which not only cause a production of drug-resistant bacteria, but also increase the medical burden [28,29]. Therefore, avoiding the diagnostic interference caused by blood culture contamination is an urgent problem to be solved in clinic. In our research, there were eight samples contaminated with Gram-positive bacteria positive by blood culture that were negative for PGN; we speculate the reason for this was that the immune cells in the body would maintain activity for a period of time after the blood isolated, and when the contaminated bacteria entered the blood, they would be quickly swallowed and killed by the immune system of the human blood system or would live in a static state, unable to reproduce continuously. Thus, the specific polysaccharide fragments were rarely produced and were further decomposed into unrecognized fragments by lysozyme, so they would not interfere with the detection of PGN, LPS, and BDG. However, when these contaminated bacteria entered the blood culture bottle with a rich culture matrix, they would grow and propagate rapidly. Moreover, the internal environment of the blood culture bottle could also crack some immune cells and inactivate some immune factors. Therefore, the tests of PGN, LPS, and BDG might well avoid the false positive results caused by *coagulase negative Staphylococcus* and positive bacilli during clinical blood testing.

The low positive rate of blood culture is also a problem in clinical testing. At present, 28–49% of patients with severe bacterial sepsis have negative blood culture results. Research [30,31,32] has shown that the amount of blood collected has an important bearing on the positive rate of the blood culture; for each additional milliliter of blood collected, the positive rate increases by 2–4%. However, because of the patient’s age, acute physiological evaluation, and chronic health evaluation, the collection of sufficient blood samples is still a challenge in clinic. In our study, the positive rate of the multi-marker detection approach of BSIs was significantly higher than that of the blood culture (χ^2^ = 21.396, *p* < 0.05); especially for the detection of fungi, the positive rates of fungal blood culture and BDG were 1.62% and 9.13%, respectively. The reason for this could be as follows: After some patients were infected, only the biomarkers entered the blood but not the bacteria, or the bacteria did not grow and reproduce after entering the blood, thus resulting in negative results in the blood culture but positive results using the multi-marker detection approach. In addition, some patients with severe infection used broad-spectrum antibiotics for a long time, which eventually led to false negative blood culture, but did not significantly affect the detection of PGN, LPS, and BDG.

Certainly, as biomarkers in BSIs, the detection of PGN, LPS, and BDG also have some limitations. For example, some major operations would lead to false positive results owning to the introduction of exogenous pollution from large open wounds, the use of gauze, long operation times, and exogenous blood transfusion [33]. Clinical plasma or blood cell infusion also might lead to a false positive. In addition, the introduction of exogenous PGN or its analogues during bedside hemofiltration might lead to a false positive. The excessive use of heparin could result in a false negative, because the high concentration of heparin has an inhibitory effect on the detection system. Finally, the detection rates of PGN, LPS, and BDG were low in the samples with a positive sterile body fluid culture but negative blood culture—the reason for this might be because the tissue mucosal damage caused by local infection was not serious, and the markers did not entered or the blood less. Therefore, in order to improve the accuracy of the test results, we suggest that samples with the above conditions should be analyzed in combination with clinical conditions, or they should be continuously monitored.

In contrast with previous studies, our study is more likely to consider the combined detection effect of the multi-marker on bacterial and fungal infections in BSIs. Although the study of [17] showed that PGN can be used for the detection of bacterial infection, the data of peptidoglycan in clinical application are still lacking in other studies at present. BDG and LPS are currently being used as biomarkers in clinic. Previous studies [34,35] have shown that the sensitivity and specificity of BDG are 75–80% and 63–85%, respectively. Clinical studies of endotoxin in China have shown that the diagnostic sensitivity and specificity of LPS are 56.4–78.1% and 73.8–91.7%, respectively, in patients with bloodstream infection. In our study, the sensitivity of BDG and LPS were basically consistent with previous reports, and the specificity was slightly higher. The possible reasons for this are the heterogeneity of the population and the change of the operation—the use of automatic instruments can avoid some of these risks.

## 5. Conclusions

In conclusion, our study showed that the results of the detection of PGN, LPS, and BDG had good consistency with that of the blood culture, combined with clinical diagnosis. And the multi-marker detection approach had high automation and short detection time by GLP-F300 automatic enzyme labeling analyzer, which made up for the defects of the existing detection technology of BSIs to a certain extent, and provided more references for the diagnosis and treatment of clinical BSIs. It could be used for rapid and comprehensive detection of Gram-positive bacteria, Gram-negative bacteria and fungi in patients with suspected BSI or unexplained fever, and then better instructed reasonable application antibiotics in clinic.

## 6. Future Perspective

Blood culture, as the gold standard for the diagnosis of BSI, has its disadvantages, with rapid diagnosis being difficult to achieve. The detection of PGN, LPS, and BDG can quickly distinguish Gram-positive bacteria, Gram-negative bacteria, and fungi in BSIs within 3 h. We believe that with the rise in various emerging technologies, such as PCR/ESI MS and microfluidic technology, and the deepening of research, more rapid and accurate diagnoses of BSIs will be realized in the future.

## Figures and Tables

**Figure 1 diagnostics-12-01461-f001:**
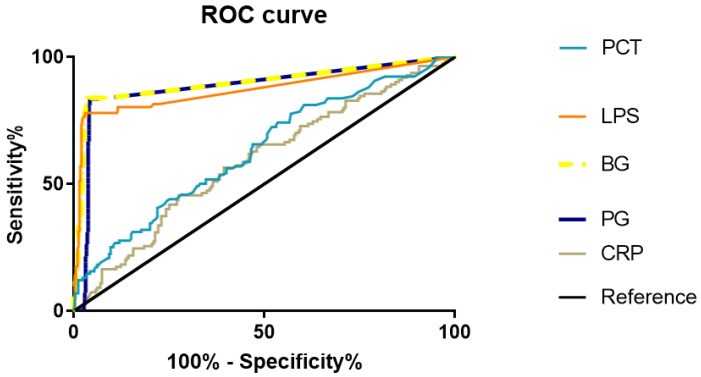
Receiver operating characteristic (ROC) curve analyses of the PGN, LPS, BDG, CRP, and PCT assays for the diagnosis of BSI (the diagnostic standard: blood culture combined with clinical diagnosis).

**Table 1 diagnostics-12-01461-t001:** The age and sex of the patients.

Factors	Patients with Definite BSI by Blood Culture Combined with Clinical Diagnosis	BSI Negative Patients	Multi-Marker Positive Patients (PGN/LPS/BDG)	Multi-Marker Negative Patients (PGN/LPS/BDG)	*p*-Value
Age (±SD), years	58.76 ± 16.77	51.97 ± 19.73			<0.001
Sex (male/female)	86/60	186/161			0.321
Age (±SD), years			58.76 ± 16.77	52.08 ± 19.62	0.014
Sex (male/female)			92/69	180/152	0.563

**Table 2 diagnostics-12-01461-t002:** Results of the blood culture combined with the clinical diagnosis and the efficacy of PGN, LPS, and BDG in the diagnosis of BSIs.

Blood Culture Combined with Clinical Diagnosis	PGN			LPS			BDG		
	+	−	Total	+	−	Total	+	−	Total
+	35	7	42	67	19	86	31	6	37
−	20	431	451	20	387	407	14	442	456
Total	55	438	493	87	406	493	45	448	493

**Table 3 diagnostics-12-01461-t003:** Results of the related indexes in PGN, LPS, and BDG and the multi-marker detection approach.

	PGN >130 pg/Ml ^a^	LPS >0.09 EU/mL	BDG >64 pg/mL	Multi-Marker Detection Approach
Sensitivity	83.3% (80.0%, 86.6%) ^b^	77.9% (74.2%, 81.6%)	83.8% (80.5%, 87.0%)	81.1% (77.6%, 84.5%)
Specificity	95.6% (93.7%, 97.4%)	95.1% (93.2%, 97.0%)	96.9% (95.4%, 98.5%)	88.7% (85.9%, 91.5%)
PPV	63.6% (59.4%, 67.9%)	77.0% (73.3%, 80.7%)	68.9% (64.8%, 73.0%)	75.5% (71.7%, 79.3%)
NPV	98.4% (97.3%, 99.5%)	95.3% (93.5%, 97.2%)	98.7% (97.6%, 99.7%)	91.6% (89.2%, 94.1%)
Total Coincidence Rate	94.5% (92.5%, 96.5%)	92.1% (89.7%, 94.5%)	95.9% (94.2%, 97.7%)	86.4% (83.4%, 89.4%)
Kappa Index	0.692	0.727	0.734	0.683

^a^. Cut off value. ^b^. 95% CI. The reason the coincidence rate of multi-marker detection approach was lower than that of single test. In the case of the multi-marker detection approach, it was judged as qualified only if all the single test results were correct, and it was judged as unqualified if any item was unqualified. Sensitivity = number of true positive cases/(number of true positive cases + number of false negative cases) × 100%; specificity = number of true negative cases/(number of true negative cases + number of false positive cases) × 100%; Positive predictive value (PPV) = number of true positive cases/(number of true positive cases + number of false positive cases) × 100%; Negative predictive value (NPV) = number of true negative cases/(number of true negative cases + number of false negative cases) × 100%; Total coincidence rate = (number of true positive cases + number of true negative cases)/total number of cases × 100%.

**Table 4 diagnostics-12-01461-t004:** Positive rates of the blood culture and the multi-marker approach detection.

Method	Gram-Positive Bacteria	Gram-Negative Bacteria	Fungi	Bacteria and Fungi
Blood culture	6.49% (32/493)	12.17% (60/493)	1.62% (8/493)	19.88% (98/493)
PGN/LPS/BDG detection	11.16% (55/493)	17.65% (87/493)	9.13% (45/493)	32.86% (162/493)

**Table 5 diagnostics-12-01461-t005:** Results of the positive rate in PCT judged according to different cut off values.

Judgment Criteria of Cut Off Values	Positive Rate in Total Samples (376 Cases)	The Positive Rate in Samples (116 Cases) with Bacteria and Fungi Positive by Blood Culture Combined with Clinical Diagnosis
>0.25 ng/mL	62.0%	75.4%
>0.5 ng/mL	49.2%	59.6%
>2 ng/mL	27.7%	41.2%
>10 ng/mL	12.5%	21.0%

**Table 6 diagnostics-12-01461-t006:** Efficacy evaluation of GN, LPS, BDG, CRP, and PCT in the diagnosis of BSI.

	PGN	LPS	BDG	CRP	PCT
Cut-off Value	>163 pg/mL	>0.115 EU/mL	>66.87 pg/mL	>103.9 ng/mL	>0.295 ng/mL
Sensitivity	83.33	77.91	83.78	74.14	45.45
Specificity	95.79	97.05	96.93	46.54	72.54
AUC	0.881	0.871	0.897	0.594	0.648
95%CI	(0.814–0.948)	(0.816–0.925)	(0.825–0.969)	(0.530–0.659)	(0.587–0.708)
*p* value	<0.01	<0.01	<0.01	<0.01	<0.01

## Data Availability

The data that support the findings of this study are available from the corresponding authors (Y.Z. and Y.C.X.) upon reasonable request.

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
