# Peer review of "The Diagnostic Values of Peptidoglycan, Lipopolysaccharide, and (1,3)-Beta-D-Glucan in Patients with Suspected Bloodstream Infection: A Single Center, Prospective Study"

_diagnostics, 2022, doi:10.3390/diagnostics12061461_

Round 1

Reviewer 1 Report

1.       Please write all the bacterial and fungal genus/species names in italics.

2.       Figure1, authors should detail the figure legend including controls and statistics. Is there any control here? Authors should use the full form for ROC at the first use. Please include the statistics and significance in the figure.

3.       Authors should include a descriptive flow diagram comparing multi-marker detection for LPS, PGN, BDG, and clinical diagnosis of BSIs.

4.       Authors should include the sample details showing age, males and females. Is the multi-marker detection or clinical diagnosis affected by the patient’s age or gender? Authors should interpret these details and include some graphs (maybe a vein diagram) to represent this data.

5.       What about the possibilities of multi-marker detection for bacterial and fungal skin infections? What is the author’s opinion about this? Please discuss.

6.       What about the possibilities for the detection of non-blood stream infections, such as Tuberculosis? Please discuss.

7.       Many viruses can also enter the bloodstream and cause infection, such as Dengue, HIV, Cytomegalovirus, etc. What about the possibilities of multi-marker detection for viral infections? Is there any marker that can be used for viruses? Please discuss. 

Author Response

  1. Please write all the bacterial and fungal genus/species names in italics.

Response: Thanks for your valuable comment. We modified all the bacterial and fungal genus/species names as italics on pages 4,5,8 in the revised manuscript, which are colored in red.

  1. Figure1, authors should detail the figure legend including controls and statistics. Is there any control here? Authors should use the full form for ROC at the first use. Please include the statistics and significance in the figure.

Response: Thanks for the valuable comment. We have added the figure legend including controls and statistics, and updated the manuscript on page 7, which are colored in red.

        Fig. 1 Receiver operating characteristic (ROC) curve analyses of PGN / LPS / BDG / CRP / PCT assays for the diagnosis of BSI (the diagnostic standard : Blood culture combined with clinical diagnosis)

Table 6. Efficacy evaluation of PGN / LPS / BDG / CRP / PCT in diagnosis of BSI

PGN

LPS

BDG

CRP

PCT

Cut-off value

>163pg/mL

>0.115EU/mL

>66.87pg/mL

>103.9ng/mL

>0.295ng/mL

Sensitivity

83.33

77.91

83.78

74.14

45.45

Specificity

95.79

97.05

96.93

46.54

72.54

AUC

0.881

0.871

0.897

0.594

0.648

95%CI

(0.814-0.948)

(0.816-0.925)

(0.825-0.969)

(0.530-0.659)

(0.587-0.708)

P value

<0.01

<0.01

<0.01

<0.01

<0.01

  1. Authors should include a descriptive flow diagram comparing multi-marker detection for LPS, PGN, BDG, and clinical diagnosis of BSIs.

Response: Thanks for the valuable comment. We have provided descriptive flow diagram in an attached table.

  1. Authors should include the sample details showing age, males and females. Is the multi-marker detection or clinical diagnosis affected by the patient’s age or gender? Authors should interpret these details and include some graphs (maybe a vein diagram) to represent this data.

Response: Thanks for the valuable comment.

General information of patients

A total of 493 patients, aged 0-94 years, were included from November 2020 to March 2021. As shown in Table 1, there were significant differences in age from three markers or blood culture plus clinical diagnosis groups, indicating that old age was also one of the susceptibility factors for bloodstream infection, while gender had no significant difference.

Table 1. The age and sex for patients

Factors

Patients with definite BSI by Blood culture combined with clinical diagnosis

BSI Negative Patients

Multi-marker  Positive Patients(PGN/LPS/BDG)

 Multi-marker  Negative Patients(PGN/LPS/BDG)

P value

Age (±SD), yrs

58.76±16.77

51.97±19.73

<0.001

Sex (male/female)

86/60

186/161

0.321

Age (±SD), yrs

58.76±16.77

52.08±19.62

0.014

Sex (male/female)

92/69

180/152

0.563

We have added these analysis on page 4 in the revised manuscript, which are colored in red.

  1. What about the possibilities of multi-marker detection for bacterial and fungal skin infections? What is the author’s opinion about this? Please discuss.

Response: Thanks for the valuable comment. Skin infectious diseases are a kind of clinically common diseases caused by pathogenic microorganisms including bacterial, fungal, viral and parasites invading human skin and mucous membranes.[1-2] Its purulent infection can cause bacteremia, sepsis and sepsis, which can lead to rapid deterioration of the disease and even life-threatening.[3] So, we thought that multi-marker detection may not be necessary for the common forms of Skin infectious, such as cellulitis, subcutaneous abscesses. However, People with purulent infection which may cause bacteremia, sepsis and sepsis should be tested with multi-marker detection.

  1. Karl T Clebak, Michael A Malone. SkinInfections. Prim Care. 2018; 45(3):433-454.
  2. Giulia Gardini, Lina Rachele Tomasoni,Francesco Castelli. Parasitic skin infections: neglected diseases or just challenging for dignosis? Curr Opin Infect Dis. 2020; 33(2):121-129.
  3. Michael M J, Binnicker M J, Sheldon C, et al. A Guide to Utilization of the Microbiology Laboratory for Diagnosis of Infectious Diseases: 2018 Update by the Infectious Diseases Society of America and the American Society for Microbiology. Clinical Infectious Diseases.2018; 67(6):813-816.
  4. What about the possibilities for the detection of non-blood stream infections, such as Tuberculosis? Please discuss.

 Response: Thanks for the comment. Tuberculosis (TB) caused by Mycobacterium tuberculosis (M.tb) is a leading cause of death from a single infectious agent worldwide. For TB diagnosis, the test methods include smear microscopy, Culturing M.tb, nucleic acid amplification test(NAATs), Tuberculin skin test (TST) and Interferon-γ-Release-Assay (IGRA), etc. [1] Despite biomarkers are urgently required to detect tuberculosis, we are sorry that multi-marker detection(LPS, PGN, BDG) in this study may not be used for the detection of Tuberculosis. However, more and more biomarkers have been founded in TB, such as interferon-inducible T-cell alpha chemoattractant (I-TAC)、 I-309、monokine induced by gamma interferon (MIG)、granulysin、fibroblast activation protein (FAP)、meprin A subunit beta (MEP1B)、furin、lymphatic vessel endothelial hyaluronan receptor 1 (LYVE-1), etc. [2-3] We believe these markers might be useful for TB diagnosis in clinic.

  1. Srivastava S, Mukhopadhyay S, Abraham P R. Aptamers: An emerging tool for diagnosis and therapeutics in Tuberculosis[J]. Frontiers in Cellular and Infection Microbiology. 2021; 11:656421.
  2. Yang Q, Chen Q, Zhang M, et al. Identification of eight-protein biosignature for diagnosis of tuberculosis. Thorax.2020; 75(7):576-583.
  3. Wykowski JH, Phillips C, Ngo T, Drain PK. A systematic review of potential screening biomarkers for active TB disease. J Clin Tuberc Other Mycobact Dis.2021; 25:100284.
  4. Many viruses can also enter the bloodstream and cause infection, such as Dengue, HIV, Cytomegalovirus, etc. What about the possibilities of multi-marker detection for viral infections? Is there any marker that can be used for viruses? Please discuss. 

Response: Thanks for the valuable comment. Pathogenic microorganisms causing bloodstream infection include bacteria, fungi, viruses, parasites, etc. Multi-marker detection (LPS, PGN, BDG) in this study just used for the detection of bloodstream infection caused by bacteria and fungi. Dengue is a mosquito-borne Flavivirus infection. According to guidelines for diagnosis and treatment of dengue in China, non-structural protein 1(NS1) can be used as a specific index for early diagnosis of dengue. In addition, studies have shown interest in the use of levels of cytokines including IL-7, IL-8, IL-10, TGF-β, TNF-α, IFN-ϒ, serum chymase levels, microparticles (MPs), such as red blood cell-derived MPs (RMPs), miRNAs (hsa-miR-21-5p, hsa-miR-146a-5p, hsa-miR-590-5p, etc) or other as predictive biomarkers of dengue.[1] Laboratory tests for HIV/AIDS mainly include HIV antibody detection, PCR, CD4+T lymphocyte count based on Chinese guidelines for diagnosis and treatment of human immunodeficiency virus/acquired immunodeficiency syndrome (2021 edition). In addition, studies have showed that C-X-C chemokine receptor 5 (CXCR5), soluble programmed cell death receptor-1 (s PD-1), homocysteine (Hcy) or other might be used as biomarkers for HIV.[2-4] Human cytomegalovirus (HCMV) is a betaherpesvirus with a global seroprevalence of 60-90%. The laboratory diagnostic methods of cytomegalovirus (CMV) infection include serological examination, histopathological examination, pp65 antigen detection, PCR, CMV IgG and IgM antibodies, etc. In addition, some domestic researches showed that IL-27, IL-37 or other may be as biomarkers for CMV.[5]

  1. Bhatt P, Sabeena S P, Varma M, et al. Current Understanding of the Pathogenesis of Dengue Virus Infection. Current Microbiology. 2021; 78(1):17-32.
  2. Ran H, Hou S, Cheng L, et al. Follicular CXCR5-expressing CD8 T cells curtail chronic viral infection. 2016;537(7620):412-428.
  3. Maike S,D Robert, Ujjwal N. Immune Checkpoints as the Immune System Regulators and Potential Biomarkers in HIV-1 Infection[J]. International Journal of Molecular Sciences. 2018; 19(7):2000.
  4. Erika, Coria-Ramirez, Leopoldo, et al. Effect of highly active antiretroviral therapy on homocysteine plasma concentrations in HIV-1-infected patients. Journal of Acquired Immune Deficiency Syndromes.2010; 54(5):477-81.
  5. Fulkerson HL,Nogalski MT, Collins-McMillen D, et al. Overview of Human Cytomegalovirus Pathogenesis. Methods Mol Biol. 2021; 2244:1-18.

Reviewer 2 Report

Manuscript presents an study aimed to assess the diagnostic values of peptidoglycan (PGN), lipopolysaccharide (LPS) and (1,3)-Beta-D-Glucan (BDG) in patients with suspected bloodstream infection.

Data are sound and methods are well performed.

Only minor revision is requiered:

"Samples which from patients met the IFI criteria of clinical diagnosis or proposed diagnosis according to relevant standard15 should be judged positive". This sentence must be explained more in detail.

The markers of PGN, LPS and 91 BDG were measured by Gram-positive bacterium Peptidoglycan Determination Assay 92 (Chromogenic), Gram-negative Bacterial Endotoxin Chromogenic Assay, Fungus (1,3)-β-93 D-glucan Chromogenic Assay and GLP-F300 automatic enzyme labeling analyzer (Au-94 tobio Diagnostics, Zhengzhou, China). Some references must be done on these techniques.

The maximum values of the abscissa and ordinate axes for the figure 1 must be corrected as it has no sense more than 100% sensitivity or specificity

Author Response

Manuscript presents an study aimed to assess the diagnostic values of peptidoglycan (PGN), lipopolysaccharide (LPS) and (1,3)-Beta-D-Glucan (BDG) in patients with suspected bloodstream infection.

Data are sound and methods are well performed.

Only minor revision is required:

"Samples which from patients met the IFI criteria of clinical diagnosis or proposed diagnosis according to relevant standard15 should be judged positive". This sentence must be explained more in detail.

Response: Thanks for the valuable comment. What we want to express is that, samples reached the level of clinical diagnosis or proposed diagnosis of IFD according to the Consensus Definitions of Invasive Fungal Disease, although without definite evidence, we classify those samples as positive samples. We have added these statements on page 3 in the revised manuscript, which are colored in red.

The markers of PGN, LPS and 91 BDG were measured by Gram-positive bacterium Peptidoglycan Determination Assay 92 (Chromogenic), Gram-negative Bacterial Endotoxin Chromogenic Assay, Fungus (1,3)-β-93 D-glucan Chromogenic Assay and GLP-F300 automatic enzyme labeling analyzer (Au-94 tobio Diagnostics, Zhengzhou, China). Some references must be done on these techniques.

 Response: Thanks for the valuable comment. The kits and instruments, being used to detect the markers of PGN, LPS and BDG were developed by Autobio Diagnostics, have been approved by Henan Medical Products Administration, and the approval numbers of Gram-positive bacterium Peptidoglycan Determination Assay (Chromogenic), Gram-negative Bacterial Endotoxin Chromogenic Assay, Fungus (1,3)-β-D-glucan Chromogenic Assay and GLP-F300 automatic enzyme labeling analyzer are 20192400460, 20192400268, 20192400281, 20202221003 respectively. We have added these approval numbers in the manuscript on page 7, which are colored in red.

The maximum values of the abscissa and ordinate axes for the figure 1 must be corrected as it has no sense more than 100% sensitivity or specificity

 Response: Thanks for the valuable comment. We have revised it in the manuscript on page 7.

Fig. 1 Receiver operating characteristic (ROC) curve analyses of PGN / LPS / BDG / CRP / PCT assays for the diagnosis of BSI (the diagnostic standard: Blood culture combined with clinical diagnosis)

Reviewer 3 Report

The authors of the paper entitled „The diagnostic values of Peptidoglycan, Lipopolysaccharide and (1,3)-Beta-D -Glucan in Patients with Suspected Bloodstream Infection:A Single center, Prospective Study” are kindly requested to consider the following recommendations:

- The section „Introduction” should be improved by adding more details on the actual state-of-the art on the considered subject. Underline the main findings and highlight them compared to the results published by other authors.

- Materials and methods: describe the sampling method for the heparin anticoagulant samples. Add information about the used software for the statistical analysis.

- Carefully check the entire manuscript so that the Latin names are written in italics.

- Discussion: compare the results from the present study with similar results published elsewhere. Discuss the differences/similarities.

- Consider including a section named „Future perspective”

Author Response

The authors of the paper entitled „The diagnostic values of Peptidoglycan, Lipopolysaccharide and (1,3)-Beta-D -Glucan in Patients with Suspected Bloodstream Infection:A Single center, Prospective Study” are kindly requested to consider the following recommendations:

- The section “Introduction” should be improved by adding more details on the actual state-of-the art on the considered subject. Underline the main findings and highlight them compared to the results published by other authors.

Response: Thanks for your valuable comment. BDG as a unique component of fungal cell wall structure has been widely used in the early diagnosis of invasive fungal infections. [1] LPS is an important indicator for the diagnosis and monitoring of gram-negative bacterial infections. [2] At present, many manufacturers have relevant products detecting LPS and BDG for clinical diagnosis of gram-negative bacterial infections and IFI. Although study [3] showed that PGN can be used for the detection of bacterial infection, studies of PGN on the source of pathogenic microorganisms in blood still relatively rare and there are no relevant products, therefore, we developed PGN detection products and studied the multi-marker detection. We have added these statements on page 2 in the revised manuscript, which are colored in red.

  1. Nucci M, Barreiros G, Reis H, et al.Performance of 1, 3-beta-D-glucan in the diagnosis and monitoring of invasive fusariosis.Mycoses.2019; 62(7):570-575.
  2. Opal SM. Endotoxins and other sepsis triggers. Contrib Nephrol.2010; 167:14-24.
  3. Inada K, Takahashi K, Ichinohe S, et al. A silkworm larvae plasma test for detecting peptidoglycan in cerebrospinal fluid is useful for the diagnosis of bacterial meningitis. Microbiol Immunol.2003; 47(10):701-707.

-Materials and methods: describe the sampling method for the heparin anticoagulant samples. Add information about the used software for the statistical analysis.

Response: Thanks for the valuable comment. Venous blood was extracted into heparin anticoagulant tube according to scale and centrifuged at 3000rpm for 10 minutes for reserve use. SPSS statistical software (version 28.0, SPSS Inc) was used for analyzing the general information for patients, and graph pad 7.0 was used for the difference between the multi-marker detection approach and CRP, PCT in the diagnosis efficacy of BSIs. We have added these statements on page 3-4 in the revised manuscript, which are colored in red.

- Carefully check the entire manuscript so that the Latin names are written in italics.

Response: Thanks for your valuable comment. We have modified the Latin names as italics on pages 4,5,8 in the revised manuscript, which are colored in red.

- Discussion: compare the results from the present study with similar results published elsewhere. Discuss the differences/similarities.

Response: Thanks for your valuable comment. In contrast to previous studies, our study is more likely to study the combined detection effect of the multi-marker on bacterial and fungal infections in BSIs. Although study 1 showed that PGN can be used for the detection of bacterial infection, the data of peptidoglycan in clinical application is still lacking in other studies at present. BDG and LPS are currently being used as biomakers in clinic, Previous studies showed that the sensitivity and specificity of BDG are 75-80%, 63-85% respectively, clinical studies of endotoxin in China showed that the diagnostic sensitivity and specificity of LPS separately are 56.4-78.1% and 73.8-91.7% in patients with bloodstream infection. In our study, the sensitivity of BDG and LPS is basically consistent with previous reports, and the specificity is slightly higher.  The possible reasons are the heterogeneity of the population and the change of the operation, the use of automatic instruments can avoid some of these risks.We have added these statements on page 9 in the revised manuscript, which are colored in red.

  1. Inada K, Takahashi K, Ichinohe S, et al. A silkworm larvae plasma test for detecting peptidoglycan in cerebrospinal fluid is useful for the diagnosis of bacterial meningitis. Microbiol Immunol 2003; 47(10):701-707.
  2. Onishi A, Sugiyama D, Kogata Y, et al. Diagnostic accuracy of serum 1,3-β-D-glucan for pneumocystis jiroveci pneumonia, invasive candidiasis, and invasive aspergillosis: systematic review and meta-analysis.J Clin Microbiol. 2012;50(1):7-15. doi:10.1128/JCM.05267-11.
  3. White SK, Walker BS, Hanson KE, Schmidt RL. Diagnostic Accuracy of β-d-Glucan (Fungitell) Testing Among Patients With Hematologic Malignancies or Solid Organ Tumors: A Systematic Review and Meta-Analysis.Am J Clin Pathol. 2019;151(3):275-285. doi:10.1093/ajcp/aqy135.

- Consider including a section named „Future perspective”

Response: Thanks for the valuable comment. Future perspective: blood culture, as the gold standard for the diagnosis of BSI, its disadvantages in rapid diagnosis are always difficult to solve. The detection of PGN, LPS and BDG can quickly distinguish Gram-positive bacteria, Gram-negative bacteria and fungi in BSIs within 3 hours. We believe that with the rise of various emerging technologies, such as PCR/ESI MS, microfluidic technology and the deepening of research, more rapid and accurate diagnosis of BSIs will be devised in the future. We have added these statements on page 10 in the revised manuscript, which are colored in red.

Round 2

Reviewer 1 Report

Authors successfully responded to the reviewer's comments and suggestions. 

Reviewer 3 Report

The authors have addressed all the comments of the reviewer. The manuscript can be accepted for publication.